# Synthesis of New Tricyclic 1,2-Thiazine Derivatives with Anti-Inflammatory Activity

**DOI:** 10.3390/ijms22157818

**Published:** 2021-07-22

**Authors:** Jadwiga Maniewska, Benita Wiatrak, Żaneta Czyżnikowska, Berenika M. Szczęśniak-Sięga

**Affiliations:** 1Department of Medicinal Chemistry, Faculty of Pharmacy, Wroclaw Medical University, Borowska 211, 50-556 Wrocław, Poland; berenika.szczesniak-siega@umed.wroc.pl; 2Department of Pharmacology, Faculty of Medicine, Wroclaw Medical University, J. Mikulicza-Radeckiego 2, 50-345 Wrocław, Poland; benita.wiatrak@umed.wroc.pl; 3Department of Inorganic Chemistry, Faculty of Pharmacy, Wroclaw Medical University, Borowska 211a, 50-556 Wrocław, Poland; zaneta.czyznikowska@umed.wroc.pl

**Keywords:** synthesis, tricyclic compounds, 1,2-thiazine, cyclooxygenase inhibition, model membrane, DSC, molecular docking

## Abstract

New, tricyclic compounds containing a sulfonyl moiety in their structure, as potential safer COX inhibitors, were designed and synthesized. New derivatives have three conjugated rings and a sulfonyl group. A third ring, i.e., an oxazine, oxazepine or oxazocin, has been added to the 1,2-benzothiazine skeleton. Their anti-COX-1/COX-2 and cytotoxic effects in vitro on NHDF cells, together with the ability to interact with model membranes and the influence on reactive oxygen species and nitric oxide, were studied. Additionally, a molecular docking study was performed to understand the binding interaction of the compounds with the active site of cyclooxygenases. For the abovementioned biological evaluation of new tricyclic 1,2-benzothiazine derivatives, the following techniques and procedures were employed: the differential scanning calorimetry, the COX colorimetric inhibitor screening assay, the MTT, DCF-DA and Griess assays. All of the compounds studied demonstrated preferential inhibition of COX-2 compared to COX-1. Moreover, all the examined tricyclic 1,2-thiazine derivatives interacted with the phospholipid model membranes. Finally, they neither have cytotoxic potency, nor demonstrate significant influence on the level of reactive oxygen species or nitric oxide. Overall, the tricyclic 1,2-thiazine derivatives are good starting points for future pharmacological tests as a group of new anti-inflammatory agents.

## 1. Introduction

Inflammation is the body’s natural response to factors that threaten homeostasis such as microbial infection or tissue damage resulting from trauma. Activation of the immune system is aimed at removing pathogens or damaged cells. Under physiological conditions, once the stimulus has been removed, the inflammation begins to cease because its initial purpose has been achieved [1]. However, fairly frequently we are dealing with prolonged inflammation lasting months or years, the so-called chronic inflammation. Many different factors lead to chronic inflammation such as failure to eliminate the initial cause of cell injury, exposure to low levels of certain irritants or foreign materials, an autoimmune disorder, a defect in the cells mediating inflammation, recurrent episodes of acute inflammation as well as molecular inducers of oxidative stress and mitochondrial dysfunction [2]. Although chronic inflammation is not a primary cause of most diseases such as cardiovascular disease, diabetes, rheumatoid arthritis, asthma, chronic obstructive pulmonary disease, chronic kidney disease, inflammatory bowel disease, cancer or Alzheimer’s disease, it contributes significantly to their pathogenesis [2,3,4,5,6,7,8,9,10,11].

There are many different inflammatory mediators such as prostaglandins, pro-inflammatory cytokines, and chemokines, but prostaglandin E_2_ (PGE_2_) is among those most important [12]. PGE_2_ is formed as a result of the transformation of arachidonic acid by the cyclooxygenase (COX). There are three isoforms of the enzyme: COX-1, COX-2, and COX-3 [13]. COX-1 is a glycoprotein that occurs under physiological conditions (constitutive form) in many organ tissues, such as the kidneys, stomach, intestines, ovaries, platelets, and fulfills many protective functions there [14]. COX-2 is expressed upon induction by various factors (e.g., cytokines, IL-1β, IL 6, TNF_α_, mitogens, growth factors) in inflammation, pain response, tissue damage or carcinogenesis. Moreover, this COX variant is produced in large amounts during the adaptive processes (e.g., at the site of the wound or ulcer healing). In recent years, the constitutive form of COX-2 has also been found in some organs, such as the spinal cord, kidneys, vascular endothelium or the uterus [14]. The COX-3 isoform is present mainly in the brain and spinal cord, and its role is not fully understood [15]. Overexpression of the inducible form of COX-2, associated with increased production of PGE_2_, plays a key role in the process of chronic inflammation [3].

The non-steroidal anti-inflammatory drugs (NSAIDs), such as aspirin, ibuprofen, meloxicam, indomethacin or nimesulide, are the best-known cyclooxygenase inhibitors. However, they show many side effects, mainly from the gastrointestinal tract (such as erosions, stomach ulcers or gastrointestinal bleeding), which get especially dangerous when NSAIDs are used over the long term in conditions associated with chronic pain and inflammation, e.g., in rheumatoid diseases. For this reason, studies are underway for the development of new, safer painkillers and anti-inflammatory drugs. 

In 2019, Rabbani published a patent review of COX-2 inhibitors. This review discusses the structures of novel COX-2 inhibitors synthesized during the last five years [16]. Our attention was drawn to the fact that the new COX-2 inhibitors differ from classic NSAIDs—they have a multi-ring structure—3-, 4- or 5-membered. This prompted us to look for second-generation COX-2 inhibitors among compounds with a polycyclic structure. As a continuation of our previous work on new oxicam derivatives, we decided to expand their molecule by adding a third ring to the 1,2-benzothiazine skeleton found in meloxicam-a classic NSAID [17]. 

Among the structures described by Rabbani, there are the sulfone derivatives obtained by El-Gamal [18]. El-Gamal synthesized a group of three-conjugated ring compounds with a sulfonyl group (Figure 1). El-Gamal’s work reassured us that our designed compounds were the right direction of research, as its sulfonyl tricyclic compounds showed promising properties—they were COX-2 inhibitors at both the enzymatic and gene levels, with a potency superior to etoricoxib, which is a selective COX-2 inhibitor. The sulfonyl group seems to enhance the anti-inflammatory effect, as many NSAIDs incorporate it in their structure, including piroxicam, meloxicam, nimesulide, celecoxib, rofecoxib or etoricoxib (Figure 2). 

Our new derivatives, similarly to the El-Gamal’s compounds, have three conjugated rings and a sulfonyl group. To the 1,2-benzothiazine skeleton in which the sulfonyl group is a part of the thiazine ring, we had added a third oxazine, oxazepine or oxazocin ring (Figure 3). The most active of El-Gamal’s compounds had also a methyl and methoxy substituent; therefore, it was planned to incorporate these substituents as well into the new structures. Besides, compounds with bromine or chlorine substituents were planned to be explored for the effects of these structural modifications on the COX-2 inhibitory activity.

Firstly, all new compounds were tested for cytotoxicity to exclude toxic compounds from further studies. The next step was to study COX-1 and COX-2 inhibition. Because the delayed phase of the inflammatory response has been associated with the neutrophil infiltration and production of free radicals and neutrophil-derived oxidants such as H_2_O_2_, O_2_˙ and HO˙, we decided to investigate the free radical scavenging ability of the new compounds [19,20].

Since COX is a membrane-bound enzyme, the ability to penetrate membranes is necessary for the drug–enzyme interactions, which was tested on the model biological membranes. Additionally, a molecular modeling study was performed to determine the binding of the new compounds to the target enzyme. In order to determine the importance of the tricyclic structure on COX activity, three bicyclic compounds with 1,2-benzothiazine scaffold were also tested.

## 2. Results and Discussion

### 2.1. Chemistry

The synthetic route of new tricyclic 1,2-thiazine derivatives is shown in Scheme 1. The conventional synthesis of the intermediates **3a**–**3e** and **4a**–**4e** was previously reported [17,21,22,23]. However, in this work, microwave synthesis, which has not been described before, was used to prepare compounds **3a**–**3e**. A mixture of saccharine **1** with appropriate 4′-substituted-2-bromoacetophenone **2a**–**2e** in dimethylformamide and triethylamine was exposed to microwave irradiation at 150 W for 3 min. The application of this method significantly shortened the reaction time from 10 h in the conventional method to 3 min. Moreover, compounds **3a**–**3e** were obtained with very high yields (95–98%). In the next step, compounds **3a**–**3e** were rearranged in *Gabriel–Colman* rearrangement, which resulted in compounds **4a**–**4e**. The final compound **5** was obtained in reaction 1-bromo-2-chloroethane with compound **4a** in acetonitrile with potassium carbonate. Compounds **6a**–**6e** were obtained in the same conditions as compound **5**, in reaction 1-bromo-3-chloropropane with appropriate compounds **4a**–**4e**. Similarly, compound **7** was obtained in reaction 1,4-dibromobutane with compound **4a**. The chemical structure and symbols of newly synthesized tricyclic 1,2-thiazine derivatives are shown in Table 1 and Scheme 1.

### 2.2. QSAR Studies

The 3D/4D QSAR model with restricted docking protocol was used to estimate the octanol–water partition coefficient (Log*P*). The lipophilicity of compounds indicates penetration of orally available drugs through biological membranes. According to the results, all compounds investigated are lipophilic. The values of the Log*P* for new tricyclic 1,2-thiazine derivatives are not exceeding 4 (Table 1). It is also worth noting that substitution of benzene ring in the case of compounds **5**–**7** lowers calculated lipophilicity of compounds. The lowest value of Log*P* was obtained for **6e** compound. The compounds with good oral absorption show Log*P* in the range from −1.0 to 5.9 [24]. The Log*P* of the newly synthesized tricyclic 1,2-thiazine derivatives falls within this range.


ijms-22-07818-t001_Table 1Table 1Chemical structure and symbols of new tricyclic 1,2-thiazine derivatives with Log*P* values.CompoundSymbolChemicalStructureLog*P* ^1^
**4a**


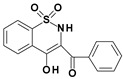

3.43
**4c**


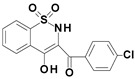

3.96
**4d**


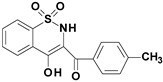

3.81
**5**


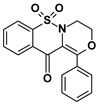

3.53
**6a**


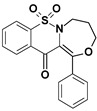

3.64
**6b**


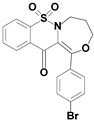

2.64
**6c**


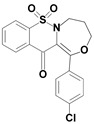

2.76
**6d**


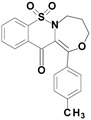

2.64
**6e**


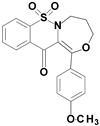

1.48
**7**


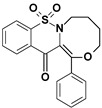

3.54^1^ The octanol–water partition coefficient (logP) predicted based on the 3D/4D QSAR algorithm Cinderella’s Shoe (CiS) for model [25].


### 2.3. Interaction with Model Membranes

To investigate the interaction of the tricyclic 1,2-thiazine derivatives (Table 1) with model membranes, we used the differential scanning calorimetry (DSC) method. As a model of the phospholipid membrane, the multibilayers made of 1,2-dipalmitoyl-*n*-glycero-3-phosphatidylcholine (DPPC) in a buffer solution (pH 7.4) were used. The influence of ten new 1,2-thiazine derivatives on the thermal properties of DPPC bilayers in pH 7.4 was examined. For comparison, the effect of meloxicam was also registered. The addition of the studied compounds caused the disappearance of the DPPC pretransition and concentration-dependent shift of the main transition temperature towards lower values, accompanied by a decrease in the transition peaks area and the broadening of the peaks (Figure 4).

The dependencies of the DPPC phase main transition temperature (T_m_) on the 1,2-thiazine derivatives: phospholipid molar ratio obtained for mixtures of DPPC with the studied compounds and meloxicam are shown in Figure 5. In DSC studies, meloxicam appeared to be the least effective compound. 

The dependencies of the DPPC phase main transition temperature (T_m_) and the peak transition half-height (∆T_½_) on DPPC mixed with **6e** compound or meloxicam are shown in Figure 4 and Figure 6. Whereas, regarding the dependencies of the transition peak half-height width (∆T_½_) and transition enthalpy (∆H) of the main phase transition of DPPC on the 1,2-thiazine derivatives, the phospholipid molar ratio obtained for mixtures of DPPC with the studied compounds and meloxicam are shown in the Appendix A.

Kyrikou and coworkers studied the thermotropic properties of DPPC multibilayers in the presence of oxicam derivatives (piroxicam, tenoxicam, meloxicam, lornoxicam) before [26]. They found that oxicam derivatives under consideration caused broadening of the main phase transition (ΔT_1/2_) of DPPC bilayers and lowering of the main phase transition temperature (T_m_). Similar results were obtained in our former studies [27,28].

The 1,2-thiazine derivatives added to model membranes influenced the thermotropic properties of DPPC in a concentration-dependent manner. Meloxicam, used as a reference drug, appeared to be the least effective compound in the studies of interaction with model membranes. All examined compounds decreased the main transition temperature (T_m_) of DPPC, increased transition peak width at the half-height (T_½_) by broadening the transition peaks, and decreased the enthalpy (ΔH) of DPPC main phase transition. The character of observed changes may allow to conclude that interactions between phospholipid molecules in the gel state became weaker in the presence of studied compounds, and that lipid polar heads as well as hydrocarbon chains regions were affected by the studied compounds (according to the standard interpretation of calorimetric data proposed by Jain and Wu) [29]. The results of the interaction of the 1,2-thiazine derivatives with the model membranes do not strictly correlate with the theoretically determined Log*P* values. It could be assumed that the greater the lipophilicity of the compound, the better the interaction with the model membrane might be. However, too high lipophilicity may result in the increased nonspecific plasma protein binding [24]. In our results, in case of parameter changes of DPPC gel-liquid crystalline phase transition studied here, the most pronounced effects were found in the presence of compounds **6c**, **6d** and **6e**, whose Log*P* is approximately 2 (see Table 1 and Figure 5). These compounds differ only by the substituent in benzene ring—a chlorine atom (compound **6c**), a methyl group (compound **6d**) or a methoxy group (compound **6e**). The difference of the interaction of compound **6e** and the reference drug (meloxicam) with model membranes formed from DPPC is shown in Figure 4. The effect is much more pronounced for compound **6e** than for meloxicam.

### 2.4. Biological Tests 

#### 2.4.1. Viability of Cell Cultures

The viability of the Normal Human Dermal Fibroblasts (NHDF) cells was assessed after 24 h incubation with 10, 50 and 100 µM tested compounds in MTT assay. A concentration-dependent decrease in cell viability was observed for all compounds tested (Figure 7). After incubation with **4a**, **4c** and **6c** compounds at 10 µM and 50 µM, a statistically significant increase in mitochondrial activity (possibly related to an increase in proliferation) was noted. In **4a** and **4c**, no cytotoxic effect was observed in the entire concentration range tested. The less than 30% decrease in culture viability was observed only in concentration of 100 µM for compound **6b** in comparison to control (i.e., no cytotoxic potential of tested compound). Results of viability of cell cultures test showed that all new 1,2-thiazine derivatives are non-toxic for a NHDF cell line and have no cytotoxic potencies. Thus, all new compounds were directed to further research.

#### 2.4.2. The COX Colorimetric Inhibitor Screening Assay

The impact of three bicyclic and seven tricyclic 1,2-thiazine derivatives, and meloxicam, as reference drug, on COX-1 and COX-2 activities was tested. The IC_50_ values were calculated separately for COX-1 and COX-2 activity estimations at 2 min of incubation with the tested compounds. The selectivity of the compounds for COX-1 or COX-2 was assessed by calculation of the IC_50_ ratios. Results of COX-1 and COX-2 inhibitory activity for all studied compounds and meloxicam are given in Table 2.

All compounds tested—both bicyclic (**4a**, **4c** and **4d**) and tricyclic (**5**, **6a**–**6e** and **7**)—inhibited COX-1 and COX-2 activity (Table 2). Moreover, most of the tested compounds (except **6b** and **6c**) showed preferential inhibition of COX-2 compared to COX-1. The **4a**, **4c**, **4d**, **5**, **6e** and **7** compounds had stronger COX-2 inhibition effect than the meloxicam. For **4a**, **4c**, **4d**, **6e** and **7** compounds were observed higher COX-2/COX-1 selectivity than for meloxicam, considered to be the preferred COX-2 inhibitor. However, also for the **5**, **6a** and **6d** compounds, this ratio is close to the reference one, which shows that most of the compounds studied are preferential COX-2 inhibitors, similarly to meloxicam. 

Although compounds **4a**, **4c** and **4d** have different chemical structure than **5**, **6e** and **7** compounds, because they have no third conjugated ring, they all showed similar high inhibitory activity towards COX-2. This suggests that the third ring, attached to the 1,2-benzothiazine skeleton, is not essential for this activity. Comparing the **4a**, **4c** and **4d** compounds with each other, it can be seen that they show very similar activity, which indicates that the substituent on the phenyl ring (H, Cl or CH_3_) is irrelevant to this activity in this group of compounds.

In the tricyclic group of 1,2-thiazine derivatives, most of the compounds with the oxazepine ring (**6a**–**6d**) showed weaker activity than the derivative containing the ring of oxazine (**5**) and oxazocin (**7**), suggesting that the ring size is important for enzyme inhibition. However, the highest COX-2 selectivity showed **6e** compound, with oxazepine ring and methoxy substituent, indicating that the effect of the substituent on the inhibitory potency of COX-2 is also significant. However, a full understanding of the structure–activity relationship (SAR) in the group of tricyclic 1,2-thiazine derivatives requires further in-depth studies. 


ijms-22-07818-t002_Table 2Table 2IC_50_ values calculated for COX-1 and COX-2 enzymes after incubation for 2 min with the 1,2-thiazine derivatives or meloxicam, and COX selectivity ratio (mean (SD); n = 3).CompoundIC50 [µM]Ratio:COX-2/COX-1COX-1COX-2
**4a**
91.2 (0.18)54.6 (0.06)0.6
**4c**
89.0 (0.16)55.9 (0.08)0.63
**4d**
89.3 (0.09)55.3 (0.10)0.62
**5**
66.1 (0.08)56.9 (0.13)0.86
**6a**
105.5 (0.15)89.9 (0.14)0.85
**6b**
86.9 (0.15)95.1(0.10)1.09
**6c**
89.8 (0.14)94.1 (0.12)1.05
**6d**
125.7 (0.07)92.9 (0.09)0.74
**6e**
115.3 (0.11)56.9 (0.06)0.49
**7**
86.1 (0.08)54.0 (0.09)0.63meloxicam83.7 (0.10)59.2 (0.12)0.71SD values are given in brackets.


#### 2.4.3. Reactive Oxygen Species (ROS) and Nitric Oxide (NO)

The level of reactive oxygen species (ROS) was evaluated in DCF-DA assay and nitric oxide (NO) in Griess assay, on NHDF cells. The tested compounds did not cause a significant increase or decrease in the level of free oxygen radicals (ROS), except for compound **4a** at a concentration of 10 µM, when a statistically significant increase was observed compared to the control (Figure 8). This increase may be due to the normal cellular activity due to the increased cell numbers, probably resulting from proliferation. 

Similarly as in ROS studies, all compounds tested showed no increase or decrease in NO level across the range of concentrations tested (data not shown). The mean level of NO ranged from 0.98 to 1.03 compared to the control group (1.0). 

### 2.5. Molecular Docking

The binding mode of the 1,2-thiazine derivatives to the binding site of both cyclooxygenases was determined by using molecular docking. All data are presented in the Appendix A and Figure 9. The crystallographic data, despite the structural diversity, exhibited that COX-1 and COX-2 enzymes have almost identical molecular weight and are characterized by similar binding site [30]. Due to their amino acid sequence homology close to 65%, the binding manner of the ligands is slightly different. The subtle differences are responsible for selectivity of inhibitors and are adopted in the design process. The formation of additional binding pockets including Leu352, Ser353, Tyr355, Phe518 and Val523 of COX-2 is connected with the replacement of Ile523 with smaller side-chain Val523 residue and the conformation changes of Tyr355. 

It is well known that scoring functions which are used in the docking algorithms only give approximate values of binding energies. Hence, it was necessary to validate them with in vitro measurements. According to the results of molecular modelling study, all compounds tested can bind to the active center of COX-2. The colorimetric inhibitor screening assay indicated that compounds **4a**, **4c**, **4d**, **5**, **6a**, **6d**, **6e** and **7** had stronger COX-2 inhibition effect than meloxicam. In the mentioned cases, the binding manner was dependent on the structural properties of docked ligand. Most of the considered ligands can bind similarly to the substrate arachidonic acid in the active site of cyclooxygenase typically through H-bonding interactions with Arg120, Tyr355 and Ser350 [31,32,33]. The analogous mode of binding is observed in the case of classic nonsteroidal anti-inflammatory drugs (NSAIDs) including indomethacin and diclofenac [34,35]. Some compounds similar to the meloxicam bind in a hydrophobic pocket comprising Ser353, Leu384, Tyr385, Trp387, Val523 and Met522 and interact with the water molecule (**4a**, **4c**, **4d**, **5** and **6e**), [36].

In the case of the **4c**-COX-2 complex, four hydrogen bonding interactions were found with Arg120 (2.84 Å; 3.30 Å), Tyr355 (3.01 Å) and Ser530 (2.81Å). As shown in Appendix A (in Appendix A), **4c** can bind to the same binding pocket of COX-2 as meloxicam including Arg120, Tyr355, Val523, Gly526, Ala527, Ser530 and Leu531, which arises due to the conformation of Tyr 355. Similar results were obtained in the case of binding to the COX-1, although ∆G of binding has higher value (−10.0 kcal/mol). Similar to meloxicam, **4c** is located near Leu117, Arg120, Leu352, Ile523, Gly526, Ala527 and Ser530 of COX-1 and can form one hydrogen bond with Ser530 amino acid. 

As determined, the **4a** and **4d** compounds are similar to **4c** orientation in the binding site of COX-2 (see Appendix A). Two hydrogen bonds are created between oxygens of the sulfonyl group of both compounds and Arg120 and Tyr355 amino acid residues. Additionally, 1,2-benzothiazine skeleton forms hydrophobic interactions with Val116, Met522, Ala527 and Leu531. There is also H-bonding interaction, which involves oxygen atoms of the carbonyl group of tested compounds and Ser530 of COX-2. On the contrary, the **4a** and **4d** compounds’ binding to the binding pocket of COX-1 is slightly different. Firstly, you can observe two hydrogen bonds creation with Arg120 and Tyr355. Moreover, benzene and methylbenzene rings are exposed towards hydrophobic and polar amino acid residues (Leu384, Tyr385, Trp387, Met522).

The addition of a third ring to the benzothiazine moiety in the tricyclic 1,2-thiazine derivatives (compounds **5** and **6a**) affects orientation of inhibitors in the binding site of enzymes. As determined, compound **5** can form two hydrogen bonds with Arg120 and Tyr355 amino acids in the case of COX-2 binding. The benzene ring can penetrate the hydrophobic pocket formed by Leu384, Tyr385, Trp387, Met522 and Gly526. Similar binding manner was observed in the binding cavity of COX-1; however, no hydrogen bonds were formed. 

The oxazepino-benzothiazine derivatives (compounds **6b**, **6c**, **6d** and **6e**) bind to the enzymes differently than the ligands described above. The **6b** and **6c** compounds can bind more strongly to COX-1, which is proven by inhibition measurements (see Appendix A). In contrast to COX-2, COX-1 can form two hydrogen bonds with oxygen atoms of oxazepine and the sulfonyl group of **6b**. In the vicinity of the bromobenzene ring, the hydrophobic and polar amino acids Leu93, Val116, Leu359 and Tyr355 are present. The binding mode of **6c** in the active center of protein is similar; however, in this case three hydrogen bonds with Arg120, Ser530 and Leu531 of COX-1 are formed. 

The replacement of halogens with a methyl and metoxy substituent does not substantially change the binding manner in the case of binding to the COX-1 pocket. The compounds **6d**, **6e** and **6a** (without a substituent in the benzene ring) binding to the protein are exposed towards mainly hydrophobic amino acid residues (Leu93, Val116, Leu117, Ile345, Val349, Leu357, Leu359, Ile523, Ala527 and Ser530) which can be involved in the *van der Waals* type of interactions. The first two inhibitors also form three hydrogen bonds with Arg120, Ser530, Leu531 of COX-1. 

In the case of interactions with COX-2, **6a** and **6d** interact via two hydrogen bonds with Arg120 moiety with a distance between electronegative atoms close to 3 Å. The way of binding is typical for the inhibitors with oxazepine-benzothiazine moiety (**6a**–**6d**). 

According to the enzymatic measurements, the **6e** compound is the most selective inhibitor of COX-2 (the lowest value of the COX-2-/COX-1 ratio). As can be observed in Figure 9, compound **6e** exhibits a unique binding configuration. As can be seen, in this case the oxazepine-benzothiazine moiety occupied the characteristic hydrophobic pocket of COX-2 formed by Val349, Leu352, Tyr355, Ser353, Leu359, Tyr385, Trp387, Val523, Gly526 and Ala527 and forms a hydrogen bond with water molecule (H_2_O784 (2.96 Å)) as meloxicam. On the other hand, the methoxybenzene ring penetrates the cavity created by Ser353, Leu531, Met535 and Leu534. 

The oxazocin derivative (compound **7**) is almost located in the same binding place of COX-1. As can be observed, two hydrogen bonds are created between the oxygens of the sulfonyl group of compound **7** and Ser530 and Leu531 amino acid residues of COX-2. The hydrophobic interactions with Met113, Val116, Leu117, Arg120, Ile345, Tyr355, Leu359, Val523, Ala527 and Ser530 were also found.

## 3. Materials and Methods

### 3.1. Chemistry

The reagents and solvents were obtained from commercial sources (Merck Life Science, Merck KGaA, Darmstadt, Germany; Sigma-Aldrich, Merck KGaA, Darmstadt, Germany, Alfa Aesar, Thermo Fisher GmbH, Kandel, Germany). All the chemicals were of analytical grade and used without further purification. Melting points were determined in open glass capillaries using a MEL-TEMP melting-point apparatus and were uncorrected. Thin-layer chromatography (TLC) was performed on aluminum sheets pre-coated with Merck silica gel 60 F254, and detection was achieved under ultraviolet (UV) light. ^1^H and ^13^C NMR spectra were recorded on a Bruker 300 MHz spectrometer using CDCl_3_ as a solvent. Chemical shifts for proton nuclear magnetic resonance (^1^H NMR) spectra are reported in parts per million (ppm) relative to the signal of tetramethylsilane at 0 ppm (internal standard). Splitting patterns are designated as follows: s, singlet; t, triplet; m, multiplet. Chemical shifts for carbon nuclear magnetic resonance (^13^C NMR) spectra are reported in parts per million (ppm) relative to the center line of the CDCl_3_ triplet at 76.9 ppm. FT-IR spectra were recorded on a Perkin-Elmer Spectrum Two UATR FT-IR spectrometer. Mass data were acquired on a Bruker Daltonics micrOTOF-Q Mass Spectrometer in a positive ion mode with flow-injection electrospray ionization (ESI). The elemental analyses were carried out on a Carlo Erba NA 1500 analyzer and were within ±0.4 % of the theoretical value. 

#### 3.1.1. The Microwave Synthesis of Compounds **3a**–**3e**

The conventional synthesis and experimental data of compounds **3a**–**3e** and **4a**–**4e** were previously reported [17,21,22,23].

A mixture of commercially available saccharine (0.92 g, 5 mmol) with 5 mmol of corresponding 4′-substituted-2-bromoacetophenone **2a**–**2e** (2-bromoacetophenone for **3a**; 2,4′-dibromoacetophenone for **3b**; 4′-chloro-2-bromoacetophenone for **3c**; 4′-methyl-2-bromoacetophenone for **3d**; 4′-methoxy-2-bromoacetophenone for **3e**) in 7 mL of dimethylformamide (DMF) and 0.7 mL of triethylamine (TEA) was exposed to microwave irradiation at 150 W for 3 min. Then, the mixture was poured over ice-cooled water (50 mL), resulting in the formation of a white solid, which was filtered and washed with cold water. The solid was dried and recrystallized from ethanol to give 2-(4-substitutedphenacyl)-2*H*-benzisothiazol-3-on 1,1-dioxides with 95–98% yield. 

In the next step, compounds **3a**–**3e** were rearranged in *Gabriel–Colman* rearrangement, which resulted in compounds **4a**–**4e** [17,21,22,23].

#### 3.1.2. Synthesis and Experimental Data of Tricyclic 1,2-Benzothiazine Derivatives (**5**, **6a**–**6e** and **7**)

In a 100 mL round-bottom flask equipped with a reflux condenser and a magnetic stirrer, 3 mmol of compound **4a** (for final compounds **5**, **6a** and **7**) or compound **4b** (for **6b**), compound **4c** (for **6c**), compound **4d** (for **6d**) or compound **4e** (for **6e**) were dissolved in 10 mL of acetonitrile, and then 9 mmol of 1-bromo-2-chloroethane (for compound **5**) or 1-bromo-3-chloropropane (for compounds **6a**–**6e**) or 1,4-dibromobutane (for compound **7**) and 9 mmol of anhydrous potassium carbonate were added. The obtained suspension was stirred at reflux for 5 h. When the reaction ended, which was controlled on TLC plates, acetonitrile was distilled off, the residue was treated with 50 mL of chloroform and insoluble materials were filtered off. The filtrate was then evaporated and the residue was purified by crystallization from ethanol to give desirable products **5**, **6a**–**6e** and **7** with medium yields.

##### 6,6-dioxo-1-phenyl-3,4-dihydro-[1,4] oxazino [4,3-b][1,2]benzothiazin-11-one (**5**)

Yellow powder, 56% yield, mp 182–183 °C; FT-IR (cm^−1^): 1665 (C=O), 1325, 1172 (SO_2_). ^1^H NMR (300 MHz, CDCl_3_) *δ* (ppm): 4.00–4.03 (t, *J* = 4.8 Hz, 2H, N-CH_2_), 4.53–4.56 (t, *J* = 4.8 Hz, 2H, O-CH_2_), 7.43–7.51 (m, 5H, arom.), 7.65–7.75 (m, 2H, arom.), 7.89–7.92 (m, 1H, arom.), 8.04–8.07 (m, 1H, arom.). ^13^C NMR (300 MHz, CDCl3) *δ* (ppm): 177.37, 157.55, 139.27, 134.17, 133.17, 132.97, 131.32, 130.17, 129.08, 128.96, 128.06, 121.53, 115.36, 67.46, 38.79. HRMS (ESI) calcd. for C_17_H_13_NO_4_S [M+H]+ 328.0638; found: 328.0628. Anal. calcd. for C_17_H_13_NO_4_S: C, 62.37; H, 4.00; N, 4.28; found: C, 62.83; H, 4.05; N, 4.21.

##### 7,7-dioxo-1-phenyl-4,5-dihydro-3H-[1,4]oxazepino[4,3-b][1,2]benzothiazin-12-one (**6a**)

Beige powder, 52% yield, mp 171–172 °C; FT-IR (cm^−1^): 1698 (C=O), 1352, 1177 (SO_2_). ^1^H NMR (300 MHz, CDCl3) *δ* (ppm): 1.98 (m, 1H, 4-CH_ax_), 2.22–2.27 (m, 1H, 4-CH_eq_), 2.62–2.73 (m, 1H, 5-CH_ax_), 3.06–3.15 (m, 1H, 5-CH_eq_), 3.51–3.60 (m, 1H, 3-CH_ax_), 4.04–4.12 (m, 1H, 3-CH_eq_), 7.44–7.60 (m, 3H, arom.), 7.73–7.86 (m, 3H, arom.), 8.16–8.27 (m, 3H, arom.). ^13^C NMR (300 MHz, CDCl3) *δ* (ppm): 194.64, 189.32, 139.10, 135.06, 133.95, 133.25, 133.18, 129.71, 129.38, 128.48, 128.40, 124.56, 83.23, 50.96, 36.08, 22.81. HRMS (ESI) calcd. for C_18_H_15_NO_4_S [M+H]+ 342.0795; found: 342.0803. Anal. calcd. for C_18_H_15_NO_4_S: C, 63.33; H, 4.43; N, 4.10; found: C, 63.44; H, 4.15; N, 4.49.

##### 1-(4-bromophenyl)-7,7-dioxo-4,5-dihydro-3H-[1,4]oxazepino[4,3-b][1,2]benzothiazin-12-one (**6b**)

Beige powder, 39% yield, mp 188–189 °C; FT-IR (cm^−1^): 1706 (C=O), 1344, 1177 (SO_2_). ^1^H NMR (300 MHz, CDCl_3_) *δ* (ppm): 1.97 (m, 1H, 4-CH_ax_), 2.26 (m, 1H, 4-CH_eq_), 2.59–2.64 (m, 1H, 5-CH_ax_), 3.08–3.10 (m, 1H, 5-CH_eq_), 3.53–3.56 (m, 1H, 3-CH_ax_), 4.05–4.08 (m, 1H, 3-CH_eq_), 7.60–7.62 (m, 2H, arom.), 7.76–7.86 (m, 3H, arom.), 8.04–8.07 (m, 2H, arom.), 8.24–8.27 (m, 1H, arom.). ^13^C NMR (300 MHz, CDCl3) *δ* (ppm): 193.58, 188.90, 138.92, 135.17, 133.27, 132.63, 131.84, 131.23, 129.43, 128.54, 128.27, 124.62, 83.14, 50.90, 35.95, 22.76. HRMS (ESI) calcd. for C_18_H_14_BrNO_4_S [M+H]+ 419.9900; found: 419.9906. Anal. calcd. for C_18_H_14_BrNO_4_S: C, 51.44; H, 3.36; N, 3.33; found: C, 51.68; H, 3.45; N, 3.27.

##### 1-(4-chlorophenyl)-7,7-dioxo-4,5-dihydro-3H-[1,4]oxazepino[4,3-b][1,2]benzothiazin-12-one (**6c**)

Beige powder, 19% yield, mp 168–170 °C; FT-IR (cm-1): 1709 (C=O), 1348, 1174 (SO_2_). ^1^H NMR (300 MHz, CDCl3) *δ* (ppm): 1.99 (m, 1H, 4-CH_ax_), 2.28 (m, 1H, 4-CH_eq_), 2.60–2.64 (m, 1H, 5-CH_ax_), 3.08–3.11 (m, 1H, 5-CH_eq_), 3.53–3.56 (m, 1H, 3-CH_ax_), 4.05–4.08 (m, 1H, 3-CH_eq_), 7.42–7.45 (m, 2H, arom.), 7.76–7.85 (m, 3H, arom.), 8.12–8.15 (m, 2H, arom.), 8.24–8.26 (m, 1H, arom.). ^13^C NMR (300 MHz, CDCl_3_) *δ* (ppm): 193.38, 188.92, 139.75, 138.92, 135.17, 133.27, 132.19, 131.17, 129.43, 128.84, 128.27, 124.62, 83.14, 50.90, 35.95, 22.75. HRMS (ESI) calcd. for C_18_H_14_ClNO_4_S [M+H]+ 376.0405; found: 376.0401. Anal. calcd. for C_18_H_14_ClNO_4_S: C, 57.52; H, 3.75; N, 3.73; found: C, 57.60; H, 3.50; N, 3.73.

##### 7,7-dioxo-1-(p-tolyl)-4,5-dihydro-3H-[1,4]oxazepino[4,3-b][1,2]benzothiazin-12-one (**6d**)

Beige powder, 46% yield, mp 204–206 °C; FT-IR (cm-1): 1701 (C=O), 1336, 1175 (SO_2_). ^1^H NMR (300 MHz, CDCl_3_) *δ* (ppm): 1.97 (m, 1H, 4-CH_ax_), 2.55 (m, 1H, 4-CH_eq_), 2.43 (s, 3H, CH_3_), 2.68–2.72 (m, 1H, 5-CH_ax_), 3.09–3.15 (m, 1H, 5-CH_eq_), 3.54–3.60 (m, 1H, 3-CH_ax_), 4.06–4.11 (m, 1H, 3-CH_eq_), 7.28–7.29 (m, 2H, arom.), 7.76–7.86 (m, 3H, arom.), 8.09–8.12 (m, 2H, arom.), 8.25–8.28 (m, 1H, arom.). ^13^C NMR (300 MHz, CDCl3) *δ* (ppm): 194.23, 189.40, 144.20, 139.14, 134.99, 133.11, 131.21, 129.84, 129.34, 129.19, 128.41, 124.53, 83.20, 50.96, 36.02, 22.74, 21.72. HRMS (ESI) calcd. for C_19_H_17_NO_4_S [M+H]+ 356.0951; found: 356.0976. Anal. calcd. for C_19_H_17_NO_4_S: C, 64.21; H, 4.82; N, 3.94; found: C, 64.21; H, 4.69; N, 3.89.

##### 1-(4-methoxyphenyl)-7,7-dioxo-4,5-dihydro-3H-[1,4]oxazepino[4,3-b][1,2]benzothiazin-12-one (**6e**)

Beige powder, 50% yield, mp 144–145 °C; FT-IR (cm^−1^): 1709 (C=O), 1338, 1173 (SO_2_). ^1^H NMR (300 MHz, CDCl3) *δ* (ppm): 1.97 (m, 1H, 4-CH_ax_), 2.26 (m, 1H, 4-CH_eq_), 2.71 (m, 1H, 5-CH_ax_), 3.16 (m, 1H, 5-CH_eq_), 3.57 (m, 1H, 3-CH_ax_), 3.87 (s, 3H, OCH_3_), 4.09 (m, 1H, 3-CH_eq_), 6.92–6.96 (m, 2H, arom.), 7.74–7.82 (m, 3H, arom.), 8.18–8.26 (m, 3H, arom.). ^13^C NMR (300 MHz, CDCl3) *δ* (ppm): 193.09, 189.42, 163.57, 139.06, 134.98, 133.11, 132.16, 129.33, 128.40, 126.44, 124.52, 113.72, 83.13, 55.48, 50.96, 36.96, 22.64. HRMS (ESI) calcd. for C_19_H_17_NO_5_S [M+H]+ 372.0900; found: 372.0897. Anal. calcd. for C_19_H_17_NO_5_S: C, 61.44; H, 4.61; N, 3.77; found: C, 61.02; H, 4.63; N, 3.45.

##### 8,8-dioxo-1-phenyl-3,4,5,6-tetrahydro-[1,4]oxazocino[4,3-b][1,2]benzothiazin-13-one (**7**)

Beige powder, 12% yield, mp 198–201 °C; FT-IR (cm^−1^): 1703 (C=O), 1343, 1172 (SO_2_). ^1^H NMR (300 MHz, CDCl_3_) *δ* (ppm): 1.39–1.42 (m, 1H, 5-CH_ax_), 1.83–1.95 (m, 3H, 5-CH_eq_ i 4 CH_2_), 2.13–2.23 (m, 1H, 6-CH_ax_), 2.68–2.77 (m, 2H, 6-CH_eq_ i 3-CH_ax_), 4.07–4.11 (m, 1H, 3-CH_eq_), 7.43–7.58 (m, 3H, arom.), 7.73–7.80 (m, 3H, arom.), 8.28–8.34 (m, 3H, arom.). ^13^C NMR (300 MHz, CDCl3) *δ* (ppm): 194.98, 188.81, 137.34, 135.08, 134.40, 133.34, 133.00, 130.19, 129.05, 128.62, 128.29, 124.97, 79.01, 48.19, 30.04, 23.92, 30.39. HRMS (ESI) calcd. for C_19_H_17_NO_4_S [M+H]+ 356.0951; found: 356.0947. Anal. calcd. for C_19_H_17_NO_4_S: C, 63.85; H, 5.36; N, 3.92; found: C, 63.80; H, 5.22; N, 4.06.

### 3.2. Interaction with Model Membranes

#### 3.2.1. Chemicals

Tris-EDTA buffer solution (pH 7.4) and 1,2-dipalmitoyl-*n*-glycero-3-phosphatidyl- choline (DPPC) were purchased from Sigma-Aldrich, Merck KGaA, Darmstadt, Germany. None of the compounds studied were soluble in water, so their chloroform (P.P.H. STANLAB, Lublin, Poland, analytical grade) solutions were used for calorimetric experiments.

#### 3.2.2. Differential Scanning Calorimetry (DSC)

Calorimetric measurements were performed using a differential scanning calorimeter DSC 214 Polyma (Netzsch GmbH & Co., Selb, Germany) equipped with an Intracooler IC70 (Netzsch GmbH & Co., Selb, Germany) in the Laboratory of Elemental Analysis and Structural Research (Faculty of Pharmacy, Wroclaw Medical University, Wroclaw, Poland). For each sample, 2 mg of phospholipid (DPPC) were dissolved in the appropriate amount of chloroform stock solution (5 mM) of the compounds studied (the compound: DPPC molar ratios in the samples were: 0.06, 0.08, 0.10, 0.12). The solvent was then evaporated under a stream of nitrogen gas. After that, the residual solvent was evacuated under vacuum (Rotary evaporator, Büchy Poland, Warsaw, Poland) for 2 h. In this process, the phospholipid was transferred onto the dry film on the inner surface of the Eppendorf tube. Samples were hydrated by 20 μL of Tris–EDTA buffer (pH 7.4). Hydrated mixtures of DPPC, compounds studied and buffer, closed in Eppendorf tubes, were heated (Labnet Dry Bath, Labnet International Inc.) to the temperature higher by 10 °C than the main phase transition temperature of the phospholipid used (DPPC) and vortexed (neoVortex, neoLab) until homogeneous dispersion was obtained. Then, the samples were transferred into aluminum sample pans of the Concavus^®^ type (Netzsch GmbH & Co., Selb, Germany) and sealed. A pan of the same type, filled with 20 μL of Tris–EDTA buffer (pH 7.4), was employed as a reference. Measurements of the DPPC main phase transition were performed using the heat-flow measurement method at a heating rate of 1 °C per minute over a temperature range of 30–50 °C in a nitrogen dynamic atmosphere (25 mL/min). Data were analyzed off-line using Netzsch Proteus^®^ 7.1.0 (Netzsch GmbH & Co., Selb, Germany) analysis software. The transition enthalpies were stated in [J/g]. The measured heat was normalized per gram of lipid. The apparatus was calibrated using standard samples from calibration set 6.239.2–91.3 supplied by Netzsch (Netzsch GmbH & Co., Selb, Germany). All samples were weighed on a Sartorius CPA225D-0CE analytical balance (Sartorius AG, Gottingen, Germany) with a resolution of 0.01 mg.

### 3.3. Biological Assay

#### 3.3.1. Cell Line

The study was carried out using the NHDF cell line obtained from ATCC (Manassas, VA, USA). These cells are commonly used to determine the cytotoxicity of new compounds. Cells were cultured at 37 °C in a humidified 5% CO_2_/95% air atmosphere incubator and passaged twice a week.

##### Cell Culture Media

The cells were cultivated in DMEM without phenol red supplemented with 10% fetal bovine serum (FBS), 2 mM L-glutamine, 1.25 μg/mL amphotericin B and 100 μg/mL gentamicin. Prepared culture medium was stored at 4–8 °C for up to one month.

##### Tested Compounds for Viability and ROS/NO Studies

The 1,2-thiazine derivatives were dissolved in DMSO to a stock concentration of 10 mM. All prepared stock solutions were stored at −20 °C for up to 6 months. For the experiment, the compounds tested were used in the concentration range of 10, 50 and 100 μM. To achieve the working concentrations, all compounds were dissolved in the medium, and the final DMSO concentration did not exceed 1%. The abovementioned tested compounds were tested on NHDF cells.

#### 3.3.2. Viability of Cell Cultures

The viability of the cells was assessed after 24 h incubation with 10–100 µM tested compounds in MTT assay. After incubation, cell culture with a 1 mg/mL solution of MTT for 2 h at 37 °C, the formazan crystals were dissolved in isopropanol. The plates were then shaken for 30 min, and the absorbance was measured at 570 nm using VariuScan microplate reader.

#### 3.3.3. Reactive Oxygen Species (ROS) and Nitric Oxide (NO)

Levels of reactive oxygen species (ROS) and nitric oxide (NO) were evaluated in DCF-DA (2′,7′-dichlorofluorescein diacetate) and Griess (cat. No. G7921; Thermo Fisher Scientific, Waltham, MA, USA) assays, respectively. In these assays, the cells were treated with tested compounds only for 1 h. After that time, the 50 µL of supernatant were transferred into new plates to assess the NO level. The remaining supernatant was removed, the cell culture was washed, and the 25 µM DCF-DA solution was added for 1 h at 37 °C. At the same time, the Griess reagent was added to the collected supernatant into new plates for 20 min at RT in the dark. The plates with DCF-DA solution were analyzed with the Varioskan LUX microplate reader (λ_ex_ = 485 nm and λ_em_ = 535 nm). The NO level was analyzed by the measurement of the absorbance at 548 nm using VariuScan microplate reader.

#### 3.3.4. The COX Colorimetric Inhibitor Screening Assay

The ready-to-use Cayman kit (COX Colorimetric Inhibitor Screening Assay Kit cat. No. 701050; Cayman Chemical Company, Ann Arbor, MI, USA) was applied for evaluating the COX peroxidase activity for all tested compounds. The COX colorimetric kit includes ovine COX-1 and human recombinant COX-2 isoenzymes. The compounds were dissolved in ethanol to get the final concentration of 100 µM. Each tested solution of compounds was transferred into a 96-well plate in three repetitions and then incubated for 2 min at RT with prepared reagents according to the manufacturing procedure. This assay measured the peroxidase component of COXs, which is evaluated as monitoring the oxidized form of *N, N, N’, N’*-tetramethyl-p-phenylenediamine (TMPD). The absorbance was measured using a VariuScan microplate reader at 590 nm. The data were calculated according to 3 steps. First, the average absorbance of all the samples was determined. Second, the absorbance of the background wells was subtracted from the absorbance of 100% of the starting COX-1 and COX-2 activity and the test compound wells. Finally, each sample of test compounds is subtracted from the sample with 100% initial COX-1 or COX-2 activity, then divided by the given sample by 100% initial activity and multiplied by 100 to obtain percent inhibition. The calculated value was defined as the IC_50_ (the concentration at which 50% inhibition of enzyme activity occurred for COX-1 and COX-2). The ratios of IC_50_ values (COX-2/COX-1) were calculated to determine the selectivity of inhibition of cyclooxygenases. Meloxicam was used as a reference compound because of its structural similarity to the compounds tested and relative selectivity towards COX-2.

#### 3.3.5. Statistical Analysis

All results are presented as mean ± SEM (standard error of the mean) relative to the control (E/E_0_), where E is the culture with the tested substance and E_0_ is the negative control (without tested compounds).

Statistical significance was calculated compared to the control. The normal distribution, using the Shapiro–Wilka test, was checked for all biological assays calculations. In the next step, Levene’s test was calculated to assess the equality of variances for a variable. The parametric test was also used to evaluate statistical significance for data. The *p* < 0.05 was set for significant.

### 3.4. Molecular Docking

The geometry optimization of designed compounds structures was performed at the B3LYP/6–31++G** level of theory with polarizable continuum model (PCM) including solvent effects using Gaussian 09 program [31,32,33]. Docking package AutoDock4.2 and a standard protocol were used to predict the binding mode and the free energy of binding. The semi-empirical force field includes six pair-wise evaluations of energy, and the conformational entropy lost upon binding was used. The binding affinity of ligand is directly related to the Gibbs energy of binding which can be expressed as follows:(1)$$\begin{array}{c} \Delta G=\left(V_{bound}^{ligand-ligand}-V_{unbound}^{ligand-ligand}\right)+\left(V_{bound}^{cox-cox}-V_{unbound}^{cox-cox}\right)\\ +\left(V_{bound}^{cox-ligand}-V_{unbound}^{cox-ligand}+\Delta S_{conf}\right) \end{array}$$

The V term is the sum of dispersion, hydrogen bonding, electrostatics, and desolvation energies according to the following equation:(2)V=Wvdw+∑i,j(Aijrij12−Bijrij6)+Whbond∑i,jE(t)(Cijrij12−Dijrij10)+Welec∑i,jqiqje(rij)rij+Wsol∑i,j(SiVi+SjVj)e(−rij22σ2)

The crystal structures of COX-1 (PDB ID: 4O1Z) and COX-2 (PDB ID: 4M11) with meloxicam were taken from the Protein Data Bank [35]. The polar hydrogen atoms and solvent parameters were added to the chain A of cyclooxygenases, and Gasteiger charges for each of the atoms have been assigned. The binding site was defined using a grid of 60 × 60 × 60 point with 0.375 Å spacing. The grid center was established in the active site according to crystalized inhibitor location. The validation protocol was performed by docking of meloxicam into the crystal structures of cyclooxygenases and the comparison of its position with crystal structure. Binding modes of designed compounds were visualized using Chimera with UCSF Chimera, developed by the Resource for Biocomputing, Visualization, and Informatics at the University of California and LIGPLOT v.4.5.3 programs [36,37].

## 4. Conclusions

This study presents the synthesis and biological evaluation of new 1,2-thiazine derivatives designed as new anti-inflammatory agents with potential use in pain and inflammation therapy. Their cytotoxic effects as well as anti-COX-1/COX-2 activity on NHDF cells, together with the ability to interact with model membranes and the influence on reactive oxygen species and nitric oxide, were studied. Additionally, a molecular docking study was performed to understand the binding interaction of the compounds with the active site of both cyclooxygenases. According to the results of molecular modeling and the in vitro study, most of the compounds bind more strongly to the active center of COX-2 than COX-1 (**4a**, **4c**, **4d**, **5**, **6a**, **6d**, **6e** and **7**). The examined ability of the 1,2-thiazine derivatives to penetrate lipid bilayers may indicate potential modulation of the activity of the membrane-bound cyclooxygenase. Actually, according to the results of the COX colorimetric inhibitor screening assay, all of the compounds studied demonstrated preferential inhibition of COX-2 compared to COX-1. Compound **6e** showed the highest COX-2 selectivity, and what is worth noting, higher than meloxicam, considered to be the preferred COX-2 inhibitor. Moreover, all the examined bi- and tricyclic 1,2-thiazine derivatives interacted with the phospholipid model membranes, and their calculated Log*P* ranged between 1.48 and 3.96, which may indicate high bioavailability. Finally, the new compounds demonstrate no influence on the level of reactive oxygen species or nitric oxide and have no cytotoxic potencies. Overall, the new 1,2-thiazine derivatives are good starting points for future pharmacological tests as a group of new anti-inflammatory agents.

## Data Availability

The data presented in this study are available in the article and in the Appendix A.

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
