# Peer review of "Synthesis of New Tricyclic 1,2-Thiazine Derivatives with Anti-Inflammatory Activity"

_ijms, 2021, doi:10.3390/ijms22157818_

Round 1

Reviewer 1 Report

Anti-inflammatory drugs are demanded in many therapeutic areas and COX-2 is the relevant target for this purpose. So the presented manuscript is aimed at the actual medicinal topic. In addition, the authors make a great job (synthesis of compounds, COX-2 inhibition assay, cytotoxicity study, docking study). Nevertheless, I think that this manuscript not suitable for publishing in the International Journal of Molecular Sciences.

Although the authors state that the described compounds are promising candidates for future tests as new anti-inflammatory agents, COXs inhibition assay demonstrates the opposite. Compound 7 displays the best inhibitory activity toward COX-2 (desired target) in series and its IC50 is 54.0 μM, but such value is very large. There are many COX-2 literature inhibitors with IC50 less than 1 μM (some works for example Eur. J. Med. Chem. 2014, 84, 160-172; Eur. J. Med. Chem. 2021, 209, 112918; Eur. J. Med. Chem. 2016, 108, 89-103). Moreover, IC50 of Celecoxib is 0.132 μM. The selectivity has the same issue. The best result is COX-2/COX-1 is 0.49 (2 fold; compound 6e), whereas for known inhibitors selectivity is differs in more than 10 folds.

Moreover, it is very sad that the authors did not provide copies of NMR spectra. To the best of my knowledge, these data are the principal for the presentation of new compounds.

In addition, the formulas of compounds (5, 6a-e, 7) are wrong. For example, C17H15NO4S is provided for compound 5, however, the correct formula is C17H13NO4S.

Author Response

  1. Anti-inflammatory drugs are demanded in many therapeutic areas and COX-2 is the relevant target for this purpose. So the presented manuscript is aimed at the actual medicinal topic. In addition, the authors make a great job (synthesis of compounds, COX-2 inhibition assay, cytotoxicity study, docking study). Nevertheless, I think that this manuscript not suitable for publishing in the International Journal of Molecular Sciences. Although the authors state that the described compounds are promising candidates for future tests as new anti-inflammatory agents, COXs inhibition assay demonstrates the opposite. Compound 7 displays the best inhibitory activity toward COX-2 (desired target) in series and its IC50 is 54.0 μM, but such value is very large. There are many COX-2 literature inhibitors with IC50 less than 1 μM (some works for example Eur. J. Med. Chem. 2014, 84, 160-172; Eur. J. Med. Chem. 2021, 209, 112918; Eur. J. Med. Chem. 2016, 108, 89-103). Moreover, IC50 of Celecoxib is 0.132 μM. The selectivity has the same issue. The best result is COX-2/COX-1 is 0.49 (2 fold; compound 6e), whereas for known inhibitors selectivity is differs in more than 10 folds.

Response: Thank you for this comment. As the Reviewer rightly pointed out, there are many COX-2 literature inhibitors much more selective, than our compounds. Nevertheless, we are of the opinion that less COX-2 selective compounds are also promising candidates for future tests. It is generally known that the selective COX-2 inhibitors (e.g. rofecoxib and valdecoxib) do not exhibit gastrointestinal toxicity like COX-1 inhibitors. On the other hand, selective COX-2 inhibitors in the elderly people negatively affect hypertension and cardiovascular problems. Long-term placebo-controlled studies revealed cardiovascular side effects that led to the withdrawal of rofecoxib and valdecoxib from the market in the United States and Europe (10.3390/ijms20174262; 10.7759/cureus.1144; 10.1038/ncpneuro0842).  What is more, the inhibition of COX-2 may also have a negative effect on the advancement of neurodegenerative processes. COX-2 has been shown to induce the upregulation of the glial cell line-derived neurotrophic factor – GDNF (10.1113/jphysiol.2012.249235; 10.1523/JNEUROSCI.6361-11.2013). 

On the other hand, the positive effects of selective COX-1 inhibition were demonstrated in an in vivo model. The increased accumulation of COX-1 has been identified in different pathologies of the CNS, including ischemic injury, stroke, Alzheimer's disease, and traumatic brain injury (TBI). Therefore, the pharmacological inhibition of COX-1 can be effective in reducing neuronal damage following injury (10.1590/1414-431X20143601; 10.1111/j.1460-9568.2008.06251.x). Moreover, in recent years’ research has shown that COX-1 might also be involved in the development of certain types of cancer. COX-1 and COX-2 isoforms operate in a coordinate manner in cancer pathophysiology, and COX-1 plays a pivotal role in the case of the serous ovarian carcinoma (10.3390/ph11040101; PubMed ID: 12615701). Recent reports of a possible contribution of COX-1 in analgesia, neuroinflammation, or carcinogenesis suggest that COX-1 is a potential therapeutic target for new chemical compounds. Therefore, in our opinion, it is worth looking for compounds that inhibit COX, but they do not necessarily have to be highly selective COX-2 inhibitors, especially considering the aging of the population.

  1. Moreover, it is very sad that the authors did not provide copies of NMR spectra. To the best of my knowledge, these data are the principal for the presentation of new compounds.

Response: Thank you for this remark. We are very sorry for the fact, that the Reviewers had no opportunity to check NMR spectra of our compounds. In fact, we have included them in the Supplementary materials file, to which we repeatedly refer to, in this article. However, it turned out, that this file was incorrectly uploaded on the journal's website. We apologize for the situation. We have asked the editor for help in uploading the file Supplementary materials on the journal website correctly, and hope that this way the problem will be solved.

  1. In addition, the formulas of compounds (5, 6a-e, 7) are wrong. For example, C17H15NO4S is provided for compound 5, however, the correct formula is C17H13NO4

Response: Thank you for noticing! We are grateful that you have pointed it out! The mistake has been corrected.

Reviewer 2 Report

The study of Maniewska et al., proposes Tricyclic 1,2-Thiazine derivatives as potential new anti-inflammatory compounds which is a very interesting topics since the huge need of this class of drugs. The manuscript is well written and experiments in general well-designed. However, the manuscript needs a substantial reorganization to improve the readability of the results. Hence, prior to a potential publication authors should address these following points:

  1. Rational design of this series: please better explain the rational behind the design it seems not really clear to everyone how the fusion between meloxicam and El-Gamal’s compound may be an added-value for second-generation inhibitors.

  1. Interaction with model membranes: please explain why DPPC zwiterrionic model in the introduction of this paragraph, since mixture of lipid and even with cholesterol are possible when using DSC to better model a mammalian membrane. It is not obvious that membrane penetration evidenced with this model may effectively lead to COX activity modulation.

Please replace Figures 5 and 6 by tables, this will be more suitable to evaluate values and deviations.

  1. Viability of cell cultures: why using NHDF, what is the rational? It will be rather pertinent to present in vitro Inhibitor screening assay before cytotoxicity evaluation, paragraph 2.4.2 before 2.4.1.

  1. COX colorimetric inhibitor screening assay: it is not clear from material and methods section how authors determine IC50 (Table 2, number of experiments, fitting procedure ect..). Please give more details. When examining the IC50 values that are in the same range, it is not obvious to make a real difference between the compounds and to conclude about the SAR. Authors should moderate their analysis.

  1. ROS and NO: Please explain more the rational of the results. Viability and ROS/NO results should be gathered in one part as biological evaluation. Please give more details in figure legends (exposure times, number of experiments).

  1. Molecular docking: Figure 9 is enough to explain the differential positioning of compounds with COX1 and 2 active sites, authors should put tables 3 and 4 in supplementary data. This part seems disproportionate with regards to the other experimental parts which may be problematic since the selection of hit compound have to be based on biological results. This part is rather an illustration of structural basis of inhibition and possibly can help for further optimization. Authors should be careful not to overinterpret the docking data ( for example lines 273-275).

Author Response

The study of Maniewska et al., proposes Tricyclic 1,2-Thiazine derivatives as potential new anti-inflammatory compounds which is a very interesting topics since the huge need of this class of drugs. The manuscript is well written and experiments in general well-designed. However, the manuscript needs a substantial reorganization to improve the readability of the results. Hence, prior to a potential publication authors should address these following points:

  1. Rational design of this series: please better explain the rational behind the design it seems not really clear to everyone how the fusion between meloxicam and El-Gamal’s compound may be an added-value for second-generation inhibitors.

Response: Thank you for this valuable comment. To meet your demand we have rewrote those sentences as follows:

In 2019 Rabbani published a patent review of COX-2 inhibitors. This review discusses the structures of novel  COX-2 inhibitors synthesized during the last five years [16].  Our attention was drawn to the fact that the new COX-2 inhibitors differ from classic NSAIDs - they have a multi-ring structure - 3-, 4- or 5-membered. This prompted us to look for second-generation COX-2 inhibitors among compounds with a polycyclic structure. As a continuation of our previous work on new oxicam derivatives, we decided to expand their molecule by adding a third ring to the 1,2-benzothiazine skeleton found in meloxicam - a classic NSAID [17].

Among the structures described by Rabbani there are the sulfone derivatives obtained by El-Gamal [18]. El-Gamal synthesized a group of three-conjugated rings compounds with a sulfonyl group (Fig. 1). El-gamal's work reassured us that our designed compounds were the right direction of research, as its sulfonyl tricyclic compounds showed promising properties - they were COX-2 inhibitors at both the enzymatic and gene levels, with a potency superior to etoricoxib, which is a selective COX-2 inhibitor. The sulfonyl group seem to enhance the anti-inflammatory effect as many NSAIDs incorporate it in their structure, including piroxicam, meloxicam, nimesulide, celecoxib, rofecoxib or etoricoxib (Fig. 2).

Our new derivatives, similarly to the El-Gamal’s compounds, have three conjugated rings and a sulfonyl group. To the 1,2-benzothiazine skeleton in which the sulfonyl group is a part of the thiazine ring, we had added a third oxazine, oxazepine or oxazocin ring (Fig. 3). The most active El-Gamal’s compounds had also a methyl and methoxy substituent, therefore it was planned to incorporate these substituents as well into the new structures. Besides, compounds with bromine or chlorine substituents have planned to be explored for the effects of these structural modifications on the COX-2 inhibitory activity.

  1. Interaction with model membranes: please explain why DPPC zwiterrionic model in the introduction of this paragraph, since mixture of lipid and even with cholesterol are possible when using DSC to better model a mammalian membrane. It is not obvious that membrane penetration evidenced with this model may effectively lead to COX activity modulation. Please replace Figures 5 and 6 by tables, this will be more suitable to evaluate values and deviations.

Response: Thank you for this comment. We are aware of the fact, that DPPC multi-bilayers in buffer (pH 7.4) are de facto artifitial, and very basic model of phospholipid membrane. However, it can be successfully used as a cheap and quick way to initially check whether the compounds under consideration interact with the phospholipid bilayer.

For better readability of results and faster identification of dependencies, we definitely prefer graphs to the tables with figures, but to meet your demand we have prepared 3 tables with values of DPPC phase transition temperatures (Table S1), transition half height (Table S2), and transition enthalpy changes (Table S3), and put them in the file Supplementary materials.

  1. Viability of cell cultures: why using NHDF, what is the rational? It will be rather pertinent to present in vitro Inhibitor screening assay before cytotoxicity evaluation, paragraph 2.4.2 before 2.4.1.

Response: Thank you for this comment. We wish to clarify that, according to ISO 10993 part V, the cytotoxicity assessment by MTT should be performed on fibroblast cell lines, e.g., L929. This line is of animal origin. In contrast, NHDF (normal human dermal fibroblasts) cells are fibroblasts of human origin. In our opinion, it is, therefore, one of the best cell lines for assessing the cytotoxicity of new compounds.

We would also wish to explain that, the cytotoxicity study was presented before the COX study, because, as we wrote in the Introduction, the compounds were first tested for toxicity in order to direct only non-toxic compounds to further biological research. Results of viability of cell cultures test showed that all new 1,2-thiazine derivatives are non-toxic for a NHDF cell line and have no cytotoxic potencies. Thus all new compounds were directed to further research.

  1. COX colorimetric inhibitor screening assay: it is not clear from material and methods section how authors determine IC50 (Table 2, number of experiments, fitting procedure ect..). Please give more details. When examining the IC50 values that are in the same range, it is not obvious to make a real difference between the compounds and to conclude about the SAR. Authors should moderate their analysis.

Response: Thank you for this valuable hint. Experiments for each compound tested, were performed in triplicate. The tests were carried out, and the results were prepared following the procedure described by the manufacturer:

  1. The mean absorbance of all samples was determined.
  2. The absorbance of the background wells was subtracted from the absorbance of 100% of the initial activity and the inhibitor wells.
  3. Each inhibitor sample was subtracted from the 100% initial activity sample, then the samples were divided with 100% initial activity and multiplied by 100 to obtain the percent inhibition.
  4. Finally the graphs of the percent inhibition or percent initial activity by inhibitor concentration were drawn to determine the IC50 value (concentration at which 50% inhibition occurs).

As the Reviewer rightly pointed out, when examining the IC50 values that are in the same range, it is not obvious to make a real difference between the activity of single compounds and to conclude about the SAR. For this reason, we did not draw conclusions about the structure-activity relationship with respect to specific substituents. We only tried to note some general relationships. For example, compounds 6a, 6b, 6c and 6d showed a significantly lower potency of COX-2 inhibition (with IC50 approximately 90 µM) compared to compounds 4a, 4c, 4d, 5, 6e and 7 (with IC50 approximately 50 µM). And, as we wrote, a full understanding of structure-activity relationship (SAR) in this group of compounds requires further in-depth studies at more compounds tested.

  1. ROS and NO: Please explain more the rational of the results. Viability and ROS/NO results should be gathered in one part as biological evaluation. Please give more details in figure legends (exposure times, number of experiments).

Response: Thank you for this comment. As Reviewer rightly pointed out, the viability of the cell cultures and ROS/NO studies belong to the biological evaluations, the same as COX colorimetric inhibitor screening assay. All this experiments are in one part entitled: 2.4. Biological tests. The ROS and NO studies showed that most of the tested compounds did not cause a significant increase or decrease in the level of ROS either NO. Therefore it can be concluded that these compounds will not scavenge free radicals. Three concentrations (10, 50 and 100 µM) were tested in biological tests (viability, ROS and NO). All biological tests were performed in triplicate. Within one replicate, 6 replicates were performed for each concentration. More detailed information one can find in part 3. Materials and Methods.

We also added more details in Table 2 legend, as you suggested.

  1. Molecular docking: Figure 9 is enough to explain the differential positioning of compounds with COX1 and 2 active sites, authors should put tables 3 and 4 in supplementary data. This part seems disproportionate with regards to the other experimental parts which may be problematic since the selection of hit compound have to be based on biological results. This part is rather an illustration of structural basis of inhibition and possibly can help for further optimization. Authors should be careful not to overinterpret the docking data (for example lines 273-275).

Response: Thank you for this valuable comment. To meet your demand, the following changes, based on above mentioned suggestion, were introduced to the manuscript:

  1. as suggested, Table 3 and 4 have been moved to Supplementary materials.
  2. Instead of:

According to the results of molecular modelling and in vitro study most of compounds tested bind more strongly to the active center of COX-2 (4a, 4c, 4d, 5, 6a, 6d, 6e and 7). In mentioned cases the potency of binding was similar, but the binding manner was dependent on the structural properties of docked ligand.

           We put the following passages:

It is well known that scoring functions which are used in the docking algorithms only give approximate values of binding energies. Hence, it was necessary to validate them with in vitro measurements. According to the results of molecular modelling study all compounds tested can bind to the active centre of COX-2. The colorimetric inhibitor screening assay indicated that  4a, 4c, 4d, 5, 6a, 6d, 6e and 7 had stronger COX-2 inhibition effect than the meloxicam. In mentioned cases the binding manner was dependent on the structural properties of docked ligand.

  1. Instead of:

According to the value of free energy of binding equal -12.6 kcal/mol the most stable is 4c-COX-2 complex (Table 4). In this case four hydrogen bonding interactions were found with Arg120 (2.84 Å; 3.30 Å), Tyr355 (3.01 Å) and Ser530 (2.81Å).

We put the following passages:

In the case of 4c-COX-2 complex four hydrogen bonding interactions were found with Arg120 (2.84 Å; 3.30 Å), Tyr355 (3.01 Å) and Ser530 (2.81Å).

  1. The sentence “The free energy of binding of compound 5 to the both cyclooxygenases is almost the same (-9.7 kcal/mol and -9.8 kcal/mol for complex with COX-1 and COX-2, respectively)“ was removed.
  2. Instead of:

The 6b and 6c compounds are more potent inhibitors against COX-1 which is consistent with inhibition assays (see Table 3 and 4).

We put the following passages:

The 6b and 6c compounds can bind more strongly to COX-1 what is proved by inhibition measurements (see Supplementary materials).

  1. Instead of:

The oxazocin derivative (compound 7) is almost located in the same binding place and is least effective inhibitor of COX-1 in terms of free energy of binding (-8.7 kcal/mol).

We put the following passages:

The oxazocin derivative (compound 7) is almost located in the same binding place of COX-1.

Round 2

Reviewer 1 Report

  1. I agree with the authors statement, that selective COX-1 inhibitors, also, can be useful. However, described compounds are not selective. So the presented manuscript is not provided any significant results from the viewpoint of medicinal science. I suggest submitting this work to another journal.
  2. Ok
  3. Ok

Author Response

I agree with the authors statement, that selective COX-1 inhibitors, also, can be useful. However, described compounds are not selective. So the presented manuscript is not provided any significant results from the viewpoint of medicinal science. I suggest submitting this work to another journal.

Thank you for your opinion. As we wrote in Introduction our goal was to obtain neither selective COX-2 nor selective COX-1 inhibitors, but to synthesize less toxic compounds with anti-inflammatory properties.

Recent studies show that selective COX-2 inhibitors does not resolve the problems of NSAID treatment. The primary hypothesis that COX-2 specific inhibitors are analgesic and anti-inflammatory drugs without the known toxicological problems of “classical” NSAIDs must be rejected, because of newer experiences indicating the existence of a constitutive COX-2 in different organs, interestingly also in the stomach [DOI: 10.1053/gast.1996.v111.pm8690211]. Some experiments indicated that the selective COX-2 inhibition prolonged the wound healing in chronic gastric ulcers of experimental animals. Other experimental data in COX-2 knock-out mice indicate that an intact COX-2 is essential in kidney function, in the pregnancy and important for femal fertility [DOI: 10.2174/0929867003374282].

Moreover, high selectivity of COX-2 inhibition could produce important cardiovascular side effects by upsetting the balance between TxA2 and PGI2. According to one theory, all drugs that selectively block COX-2 may have an adverse cardiovascular profile because they upset the balance between pro-thrombotic TxA2 and anti-thrombotic PGI2. Aspirin and traditional NSAIDs inhibit both TxA2 and PGI2, whereas COX-2 inhibitors decrease PGI2 production without affecting TxA2. Suppression of COX-2 dependent formation of PGI2 by coxibs may thus predispose subjects to myocardial infarction and thrombotic stroke [DOI: 10.1161/01.cir.102.8.840].

In August 2004, the results of the Therapeutic Arthritis Research and Gastrointestinal Event Trial (TARGET) were published [DOI: 10.1016/S0140-6736(04)16894-3]. TARGET was a prospective, randomized, multicenter, international study. In this large outcome study, subjects with osteoarthritis were randomized to one of two sub-studies, lumiracoxib (400 mg once daily) versus naproxen (500 mg twice daily) or ibuprofen (800 mg thrice daily). Since naproxen and ibuprofen may have differential cardioprotective effects due to their inhibition of platelet function, both drugs were used as traditional NSAID “active” controls. A total of 18,325 subjects were randomized of whom 9,117 received lumiracoxib. Observed rates of gastrointestinal ulcer complications did not differ between lumiracoxib and either NSAID group [DOI: 10.1007/1-4020-5688-5_7]. The results of this study and many others suggested that inhibiting both COX-1 and COX-2 isoform is more favorable than one of them alone.

 In conclusion, we believe that non-selective COX inhibitors are also therapeutically relevant. What is more, we believe that the publication of our research, consisting of the design, synthesis and biological evaluation of new compounds with anti-inflammatory properties, meets aims and scope of this special issue of the International Journal of Molecular Sciences.

Reviewer 2 Report

Authors have answered to most of my comments, however although they have given some details on how they determined the IC50 and performed inhibition assays they still don't provide curves for example as supplementary data. This is still not clear from the Material and methods section how these experiments "The 100 μM concentration of tested  compounds were tested according to the manufacturing procedure. Meloxicam was used  as a reference compound." (Line 558)  Please better answer this point.

Minor point : Please replace promising candidates by good starting points, it is not obvious from the biological data that these compounds may be promising in vivo candidates (Line 629). 

 "Overall, the new 1,2-thiazine derivatives are promising candidates for future pharmacological in vivo tests as a group of new anti-inflammatory agents. 

Once the authors will correct this, the manuscript will be acceptable for publication. 

Author Response

  1. Authors have answered to most of my comments, however although they have given some details on how they determined the IC50 and performed inhibition assays they still don't provide curves for example as supplementary data. This is still not clear from the Material and methods section how these experiments "The 100 μM concentration of tested compounds were tested according to the manufacturing procedure. Meloxicam was used  as a reference compound." (Line 558)  Please better answer this point.

The more detailed information on how the IC50 was calculated,  as requested by the Reviewer, were added to Materials and methods. The calculations were made according to the described 3 stages. Unfortunately, we have no curves for COX studies, as the activity was assessed only at 100 µM of studied compound with 3 replicates for each, which resulted from the high price of the test.  On the contrary, the viability of cell cultures and ROS/NO studies were performed in 3 concentrations of the tested compounds: 10, 50, and 100 μM. We did not included the curves for the viability of cell cultures and ROS/NO studies, because we did not think they provide any relevant information. We would like to apologize, because it is possible that the section Tested compounds was misleading. Thus we have revised this section by replacing it with name: Tested compounds for viability and ROS/NO studies.

Minor point : Please replace promising candidates by good starting points, it is not obvious from the biological data that these compounds may be promising in vivo candidates (Line 629).

As Reviewer suggested we have changed the sentence: Overall, the new 1,2-thiazine derivatives are promising candidates for future pharmacological in vivo tests as a group of new anti-inflammatory agents.

into:  Overall, the new 1,2-thiazine derivatives are good starting points for future pharmacological tests as a group of new anti-inflammatory agents.